# Internal Residual Strain Measurements in Carbon Fiber-Reinforced Polymer Laminates Curing Process Using Embedded Tilted Fiber Bragg Grating Sensor

**DOI:** 10.3390/polym12071479

**Published:** 2020-07-01

**Authors:** Ke-Ping Ma, Chao-Wei Wu, Yao-Tung Tsai, Ya-Chun Hsu, Chia-Chin Chiang

**Affiliations:** 1Department of Mechanical Engineering, National Kaohsiung University of Science and Technology, 415 Chien Kung Road, Kaohsiung 807, Taiwan; 1103403103@gm.kuas.edu.tw (K.-P.M.); ydung100@gmail.com (Y.-T.T.); sandy3896262@gmail.com (Y.-C.H.); 2Department of Aeronautical and Mechanical Engineering. Air Force Academy, Academy, No.Sisou 1, Jieshou W. Road, Kaohsiung 820, Taiwan; cafa95011@gmail.com; 3Department of Aviation Management, Chinese Air Force Academy, No.Sisou 1, Jieshou W. Rd., Gangshan Dist., Kaohsiung 82047, Taiwan

**Keywords:** tilted fiber bragg grating, Carbon fiber-reinforced plastics, residual strain, curing monitoring

## Abstract

Carbon fiber reinforced plastics (CFRP) have many mechanical properties that are superior to those of conventional structural materials and are becoming more and more widely used. Monitoring the curing process used to produce such composite material is important to ensure the quality of the process, especially for the characterization of residual strains after the material has been manufactured. In this study, we present a tilted fiber Bragg grating (TFBG) sensor used to monitor the curing of CFRP composite materials. The TFBG sensor was embedded into the layers of CFRP laminates to study the curing residual strain of the laminates. The experimental results showed that the curing residual stress was about −22.25 MPa, the axial residual strain was −281.351 με, and lateral residual strain of 89.91 με. The TFBG sensor was found to be sensitive to the curing residual strain of the CFRP, meaning that it has potential for use in applications involving composite curing processes. Moreover, it is indeed possible to improve the properties of composite materials via the optimization and monitoring of their curing parameters.

## 1. Introduction

Carbon fiber reinforced polymers (CFRPs) have been widely used as structural materials in the aerospace and engineering industries in recent years as they are lightweight, strong in terms of specific strength, and resistant to acids, alkali corrosion, and fatigue load [1,2,3]. CFRP products will generate residual strains during the curing stage of the process, and these residual strains cause defects before the product is in service. Fiber deformation and internal delamination are difficult to observe by the naked eye and will form a potential hazard. There is an urgent need for a convenient and low-cost method for the on-line monitoring of residual strains generated in the curing process [4,5,6,7]. The measurement of curing parameters for composite prepregs can be accomplished using conventional analytical techniques, such as differential scanning calorimetry (DSC) [8,9], dielectric analysis (DEA) [10], dynamic mechanical analysis (DMA) [11], and so on. The most commonly used of these techniques is DSC, which is used primarily to calculate the amount of heat stored in a material as it heats up (heat capacity) as well as the heat absorbed or released during chemical reactions or phase changes that the difference in the amount of heat required to increase the temperature of a sample and reference is measured as a function of temperature, with both the sample and reference maintained at a same temperature approximately throughout the experiment [12,13]. Using DSC, the properties of a resin, including its curing process, can be characterized [14]. However, this method and the others mention above are complicated and expensive, and cannot be used for real-time and in-situ shrinkage curing monitoring. On the contrary, such monitoring is necessary to improve the mechanical behavior of CFRP composites. In recent years, there have been many optical fiber sensing technologies used to monitor the curing process of CFRP composites Compared with traditional monitoring technology, these sensors have obvious advantages, due to their small size and high sensitivity. In this study, we present a new method that employs a TFBG sensor for the real-time monitoring of the curing process quality and the analysis of curing internal residual strains of CFRP. 

TFBG have a certain tilt angle between the grating plane and the fiber cross section, such that the transmission spectrum of the sensor produces many resonances, resulting in the occurrence of more complex mode coupling. Since the tilt angle and refractive index modulation determine the coupling efficiency and the bandwidth of cladding mode resonances [15,16], the transmission characteristics of TFBG provide a great amount of information related to the optical fiber and grating structures. TFBG have been widely applied in various fields, such as civil structure applications, biological/biomedical and, aerospace including temperature [17], strain [18], and concentration monitoring [19]. The TFBG sensor used in this study is small and suitable for combination with common polymer materials. It can provide a non-invasive means of real-time monitoring due to it can be easily embedded into the internal sensing area of polymer composite structure. Therefore, TFBG is suitable for application as an embedded sensor element for CFRP materials. In 2007, Buggy et al. [20] embedded a TFBG sensor in an epoxy resin to monitoring the variation of refractive index. Comparing the experimental results of the refractive index sensing with the measurements made using the technology based on optical fiber Fresnel reflection, and were found to be in agreement to within 6 × 10^−4^ RIU. In 2010, Takeda et al. [21] embedded FBG sensors into CFRP to examine the relationship between laminate thickness and internal residual strain following the curing process. The experimental results indicated that spectral distortions of the CFRP specimen following curing were caused by non-uniform axial strain on the given FBG sensor, which was attributed to the use of carbon fibers. Final spectral calculations showed an inverse relationship between laminate thickness and residual strain, and that the residual strains for the laminated layers 4, 8, and 13 were −1233 με, −1367 με, and −1650 με, respectively. 

Shen [22] embedded FBG sensors into the different laminated layers of [0°]_28_ carbon fiber epoxy composite to measure residual strain. The measurement results indicated that the residual strain for the middle layer was 370 με. Gelation and solidification of the composite material could also be observed when monitoring the curing process. In 2013, Kinet et al. [23] proposed to embed a tilted fibre Bragg grating sensor to monitoring the strain inside a composite material laminate. The loading test resulting shows the strain sensitivity about 0.021 pm/με. From the above literature survey, it is clear that most of the relevant past studies focused on measuring the mechanical properties of composite materials and evaluating any damage sustained by them, but that they lacked any consideration of curing residual strain monitoring in different layers with different angles of 0°, −45°, 90°, and 45° laminates. The aim of the current study, therefore, was to apply a TFBG sensor to monitor the characteristics of the curing process in CFRP composite materials. To that end, a TFBG sensor was embedded into the lamina of composite materials, after which the curing development as well as internal residual strain values during the curing process were measured. However, in the case of CFRP curing, residual strain can also appear, inducing modification of the cladding-mode variation. By tracking the wavelength shift of the Bragg mode and cladding mode resonance, we observed a differential evolution that was attributable to a surrounding refractive index (SRI) modification due to the polymerization of the CFRP. In this case, if there is a residual strain applied on the sensor during the curing, this strain will affect the Bragg peak shape, which is directly observed in the spectrum.

## 2. Working Principle of TFBG Sensors 

In TFBG, mode fiber coupling between the wave vectors of all vector fields occurs along the fiber axis to form a uniform axial conduction mode structure that is defined by a flat phase plane perpendicular to the guide shaft, as shown in Figure 1. The coupling occurs at resonant wavelengths, λ_Bragg_, at which the phase matching condition for an ordinary fiber grating can be expressed by the following Equation (1), is satisfied [23]:(1)$$\lambda_{\mathrm{Bragg}}=\left(n_{\mathrm{eff},\mathrm{core}}+n_{\mathrm{eff},\mathrm{core}}\right)\Lambda_{g}$$

The Bragg resonance wavelength is generated by the coupling between the forward-propagating guided mode and the backward-propagating guided mode. For TFBG, due to the tilt angle of grating surface with respect to the axis of the fiber, the grating period along fiber axis can be modified according to the following equation:(2)$$\Lambda_{g}=\frac{\Lambda }{\cos\theta }$$

Substituting Equation (2) into Equation (1) yields the following:(3)$$\lambda_{\mathrm{Bragg}}=\left(n_{\mathrm{eff}}^{\mathrm{co}}+n_{\mathrm{eff}}^{\mathrm{co}}\right)\frac{\Lambda }{\cos\theta }$$

Due to the presence of the tilted angle, part of the light propagating in a forward direction through the core mode will be coupled to the cladding mode of the backward propagation, and the resonance positions of the corresponding cladding modes change accordingly. They will be determined by the following equation [23]:(4)$$\lambda_{\mathrm{Cl},i}=\left(n_{\mathrm{eff}}^{\mathrm{co}}+n_{\mathrm{eff},i}^{\mathrm{cl}}\right)\frac{\Lambda }{\cos\theta }$$
where $n_{\mathrm{eff}}^{\mathrm{co}}$ and $n_{\mathrm{eff},i}^{\mathrm{cl}}$ are the effective indices of the fiber core mode and cladding mode, respectively, Λ is the grating period measured perpendicular to the grating planes, and θ is the tilt angle of the grating planes relative to the plane of the fiber cross section. The shift in the Bragg wavelength (red shift) occurs due to increase in effective refractive index which increases with surrounding refractive index.

TFBGs enhance the backward coupling of light from the fiber core to the cladding and are therefore sensitive to the SRI [21,22]. Contrary to the core mode resonance, these modes are indeed sensitive to SRI, as well as to temperature and strain. Equations (5) and (6) show how the Bragg wavelength and cladding modes depend on temperature, strain, and SRI variations [24].
(5)$$\frac{{\Delta \lambda }_{\mathrm{Bragg}}}{\lambda_{\mathrm{Bragg}}}=\alpha_{\mathrm{Bragg}}{\Delta T}_{+\beta_{\mathrm{bragg}}\varepsilon }$$
(6)$$\frac{{\Delta \lambda }_{\mathrm{cl},i}}{\lambda_{\mathrm{cl},i}}=\alpha_{\mathrm{cl},i}\Delta T+\beta_{\mathrm{cl},i}\varepsilon +\gamma_{\mathrm{cl},i}\Delta \mathrm{SRI}$$
where ${\Delta \lambda }_{\mathrm{Bragg}}$ and $\lambda_{\mathrm{cl},i}$ are the wavelength shifts of the Bragg and the i^th^ cladding mode resonances, respectively. The wavelength shifts occur due to variations in temperature (Δ*T*), strain (ε) and/or the surrounding refractive index (ΔSRI). The thermal coefficients ($\alpha_{\mathrm{Bragg}}$ and $\alpha_{\mathrm{cl},i}$) are roughly identical, whereas the strain coefficient $\beta_{\mathrm{Bragg}}$ is different from $\beta_{\mathrm{cl},i}$, which also depends on the considered mode (i) [24].
(7)$$\frac{\Delta \lambda \mathrm{Bragg}}{\lambda }=\left\{1-\frac{n_{\mathrm{eff}}^{2}}{2}\left[p_{12}-\upsilon \left(p_{11}+p_{12}\right)\right]\right\}\varepsilon_{z}=\left(1-p_{e}\right)\varepsilon_{z}$$
(8)$$p_{e}=\frac{n_{\mathrm{eff}}^{2}}{2}\left[p_{12}-\upsilon \left(p_{11}+p_{12}\right)\right]$$
where p_e_ is the elasto-optic coefficient and p_11_ and p_12_ represent the elasto-optic tensor, with p_11_ = 0.113, p_12_ = 0.252, n_eff_ = 1.458, and ν = 0.16. The relationship between TFBG wavelength shift and axial and parallel strain are obtained [25,26,27]. The axial residual stress, axial strain, lateral residual stress, and lateral strain in the TFBG sensor were calculated via spectrum central wavelength distances and wavelength split crest distances before and after curing.

## 3. Experimental Method

### 3.1. The Manufacturing of the TFBG Sensor

The TFBG sensor used in this study was fabricated using Germanium/Boron co-doped photosensitive fiber (Fibercore PS1250), which was manufactured using the phase mask method. First, using a wire stripper to strip buffer layer 5 cm long lengths from the photosensitive fiber, after that the alcohol was used to wipe clean the fiber. Next, the phase mask platform was rotated 10 degrees and KrF excimer laser (COHERENT; Xantos XS 500, λ = 248 nm) irradiation was applied through phase masks (Ibsen Photonics) with period of 1075.5 nm to produce the tilted grating formation. The tilted grating formation was created in the fiber core by a transverse illumination with a UV interference pattern that is formed by a pair of strong deep ultra-violet (DUV)-laser beams. An optical spectrum analyzer (OSA) was then utilized to monitor the generation of the TFBG structure in the core of the spectrum, as shown in Figure 2.

### 3.2. The Curing Monitoring Experimental Setup for Carbon Fiber-Reinforced Polymer Materials.

For the curing process, the 16 layers of thermosetting prepreg materials, which had a carbon/epoxy composition (T700/3501), were used to lay up the composite laminate at direction angle 0°, 45° and 90° in sequence with the embedded TFBG sensor. The TFBG sensor was embedded between layers 8 and 9 of the prepreg material, as shown in Figure 3. 

The curing process was performed using modified diaphragm forming (MDF) with a vacuum pump, heat chamber, and air compressor. First, a teflon fabric was laid in the mold to prevent adhesion, followed by a layer of the prepreg material, another layer of teflon fabric, and then a diaphragm and an O-ring for sealing, as shown in Figure 4. The upper mold and lower mold were connected by air compressor and vacuum pump lines and laid into a thermal chamber in order to thermoform. Figure 5 shows the experimental setup of the CFRP composite curing monitoring with TFBG sensor. An optical spectrum analyzer (OSA, Anritsu MS 9740 A) was used for measuring optical signal spectra, which has a resolution of 0.05 nm. A light source (ASE-2200, ASE light source, NXTAR Technologies Inc., Tainan, Taiwan) was attached to one end of the TFBG, while an optical spectrum analyzer was attached to the other. The temperature of the curing process was recorded by data acquisition (DAQ) card (NI-9217, National Instrument, Austin, Texas, USA). The CFRP was closely molded by the application of air pressure beyond the diaphragm and the simultaneous removal of excess gas between the laminates with vacuum pump. The curing process steps were composed of three stages. The process started with the heating stage with 76 cm/Hg vacuum pressure applied in the mod and 6 kg/cm^2^ of air pressure applied upon the diaphragm. The temperature was heated with uniform temperature from room temperature to 190 °C at a rate of 3 °C/min. The second stage was the 30-min isothermal stage, during which the temperature was consistently maintained at 190 °C. The 3^rd^ stage consisted of cooling to room temperature with natural cooling. The resin viscosity grew very high during this stage, and the resin itself had increasing stability. Moreover, residual strain occurred at the end of the stage. The curing condition parameters of the CFRP composite are shown in Figure 6.

## 4. Results and Discussion

### 4.1. DSC Analysis of CFRP Materials

The glass transition is the transition between rubbery state and glassy state. When the temperature exceeds the *T*_g_ of the CFRP, degradation of resin will occur. It possibly causes defects and reduces the workpiece strength. Base on the dilatometer method, the *T*_g_ is the turning point in the coefficient of thermal expansion curve. The DSC, as shown in Figure 7. Finally, the *T*_g_ of CFRP obtained is 138.51 °C.

### 4.2. The Result of CFRP Curing Monitoring with TFBG Sensor

The curing process was divided into three stages, namely, the heating stages, isothermal stages, and cooling stages. In the heating stage, the temperature was increased from room temperature to 190 °C. Figure 8 shows the transmission spectra of the TFBG sensor during the heating stage of the curing process. At 30 °C, a Bragg mode resonance dip was generated as a result of the pressurizing effect of the vacuum pump and air compressor, and the transmission loss of resonance dip was −2.693 dB. Throughout the heating stage, the transmission loss of spectrum exhibited little variation. The transmission loss showed increasing trend before the temperature reached 80 °C. During the heating stage, the resonance dip wavelength exhibited a red-shift trend.

Figure 9 shows the temperature dependence of wavelength and transmission loss when heating stage. The transmission loss of Bragg mode showed a gradually increasing trend after the temperature increased above 80 °C. This phenomenon occurred because the increasing temperature caused the resin to melt and coat on the TFBG. After the temperature increased above 110 °C, the resonance dip decreasing trend began to rebound because of the decreased pressure applied to the TFBG by the air compressor, vacuum pump, and resin, as the resin transformed from melted liquid into a gel state. When the temperature reached 120 °C, the increasing slope of the transmission loss was significantly larger. This phenomenon was supposedly linked to the curing cross-linking reaction. The CFRP began to solidify, causing the influence of tensile strain on the TFBG to be decreased. When the temperature increased to 170 °C, the increasing slope took a new turn and stabilized, presumably because the most intense temperature for the crosslinking reaction had been reached, after which the reaction tended to ease.

In the cooling stage, the composite was cooled from 190 °C to room temperature. Figure 10 shows the transmission spectra of the TFBG sensor during the cooling process. The Bragg mode exhibited a slow trend of shifting toward short wavelengths. When the temperature reached 30 °C, the transmission loss reached its minimum at −3.178 dB. 

Figure 11 shows the temperature dependence of wavelength and transmission loss when cooling stage. The transmission loss of Bragg mode of the overall spectrum during the cooling process was stable. After cooling to 60 °C, the wavelength was greatly red-shifted and then became stabilized after 50 °C was reached, at which point the transmission loss increased significantly. All of the above phenomena were caused by the cooling effect, which caused the lateral contraction force to begin to increase, while the lateral residual stress affected the spectrum, causing peak splits of Bragg mode.

Figure 12 illustrates the transmission spectrum before and after the curing process. After the curing process, the wavelengths were shifted toward shorter wavelengths, which may have been due to compressive residual strain, as proposed in the photoelasticity theory. There were split peaks in the TFBG sensor Bragg mode transmission spectrum. According to FBG theory, it was estimated that birefringence occurred in the TFBG sensor Bragg mode due to lateral residual strain. The split into two peaks occurred in the TFBG sensor Bragg mode due to the residual strain of the isotropic laminate. The axial residual stress, axial strain, lateral residual stress, and lateral strain in the TFBG sensor were calculated via spectrum central wavelength distances and wavelength split crest distances before and after curing (Table 1). 

## 5. Conclusions

In this paper, we successfully used TFBG with a 3° tilt angle and a diameter of 125 μm to monitor the curing of CFRP composite. The results showed that residual stress was generated from the innermost to outer most laminates (0°, −45°, 90°, and 45° isotropic laminates). Thus, it can be assumed that the material directions affect the order in which stress is generated and determines the type of strain (compression or stretching) that occurs. Double-cleavage also occurred in the TFBG sensor. It was assumed that the refractive index of the TFBG sensor changed due to anisotropic ellipse polarization caused by isotropic laminate residual stress. The experimental results showed that the TFBG sensor embedded into CFRP composite had a residual stress of about −22.25 MPa, an axial residual strain of −281.351 με, and a lateral residual strain of 89.91 με. The proposed CFRP curing residual stain monitoring technology can be applied to the manufacturing quality and damage monitoring of aerospace composites. 

## Figures and Tables

**Figure 1 polymers-12-01479-f001:**
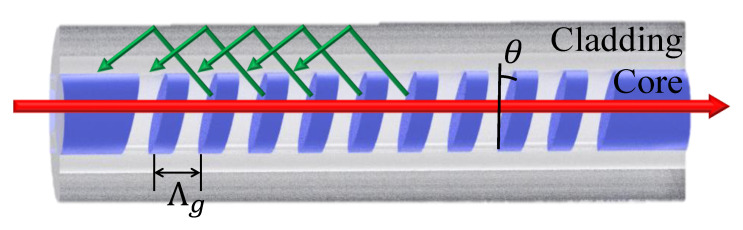
A schematic of TFBG with a tilt angle θ inscribed in the core of photosensitive fiber.

**Figure 2 polymers-12-01479-f002:**
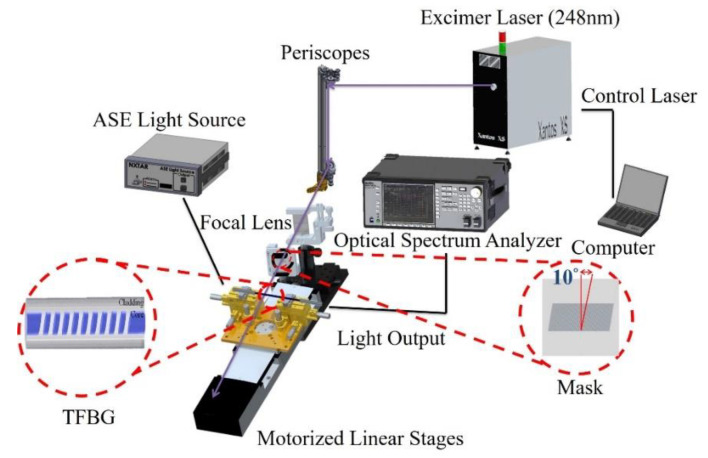
Schematic diagram of the process for TFBG fabrication.

**Figure 3 polymers-12-01479-f003:**
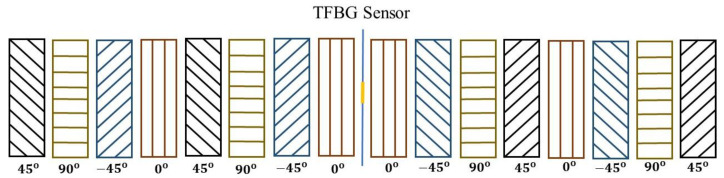
Layer directions of prepreg materials and TFBG sensor locations.

**Figure 4 polymers-12-01479-f004:**
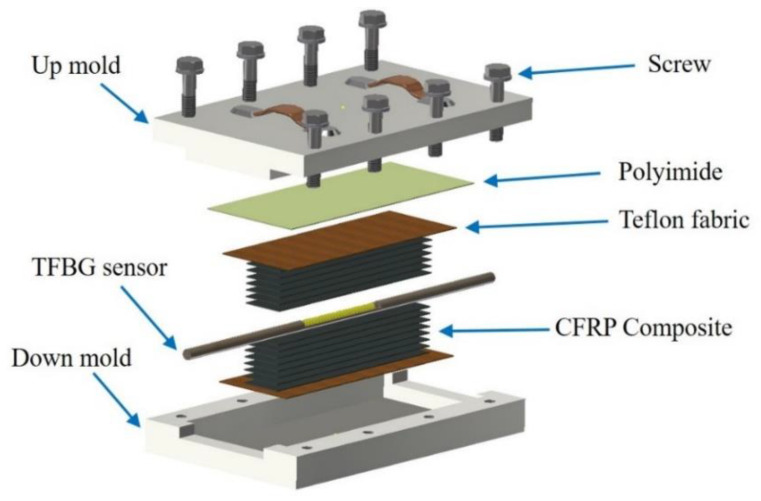
Schematic diagram of TFBG embedding and forming mold.

**Figure 5 polymers-12-01479-f005:**
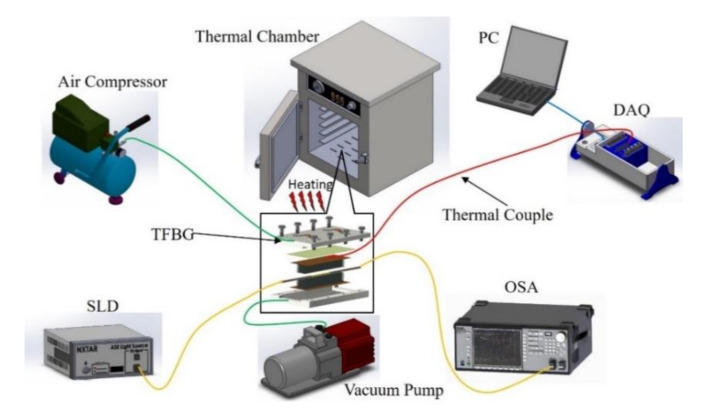
Diagram of experimental system for CFRP composite material curing monitoring with TFBG sensor.

**Figure 6 polymers-12-01479-f006:**
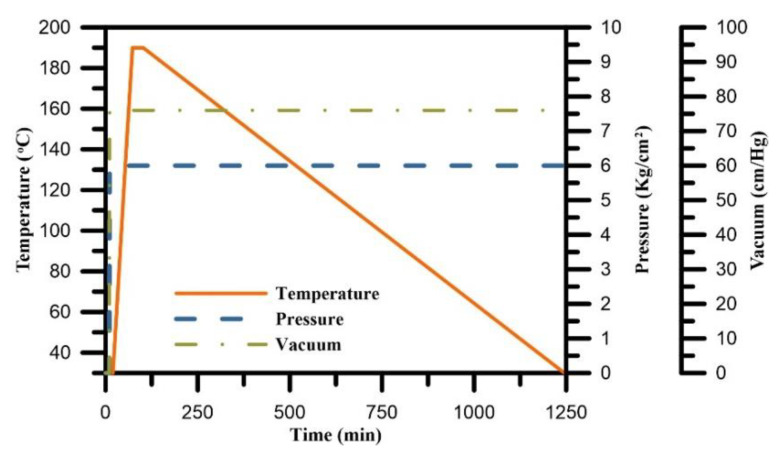
The conditions of the CFRP composite curing process.

**Figure 7 polymers-12-01479-f007:**
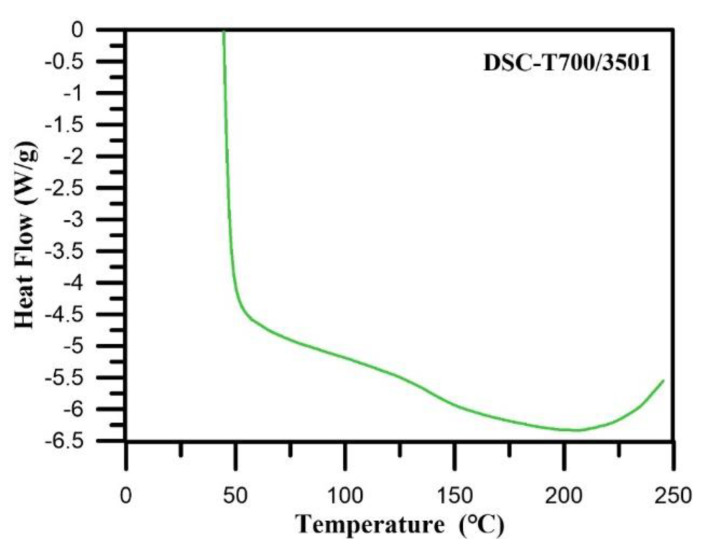
Differential scanning calorimetry (DSC) analyses of T700/3501CFRP material.

**Figure 8 polymers-12-01479-f008:**
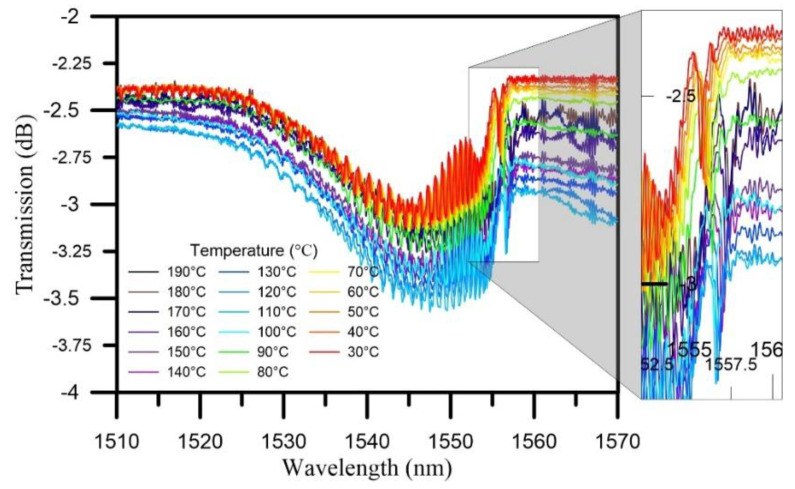
The transmission spectrum of the TFBG sensor during the heating stage.

**Figure 9 polymers-12-01479-f009:**
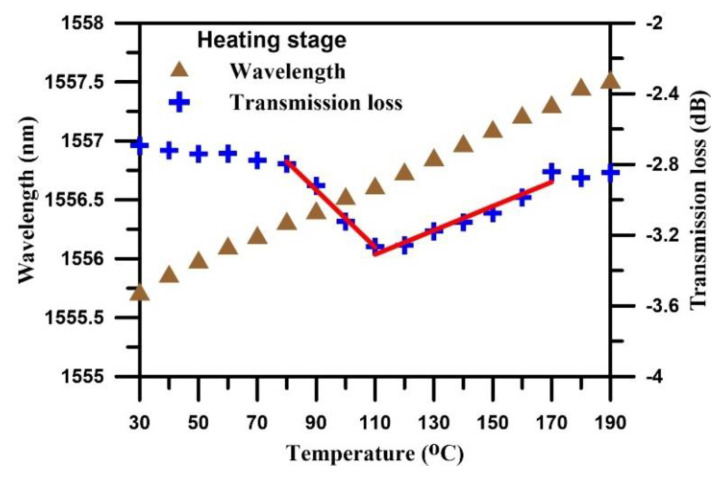
Temperature dependence of wavelength and transmission loss when heating stage.

**Figure 10 polymers-12-01479-f010:**
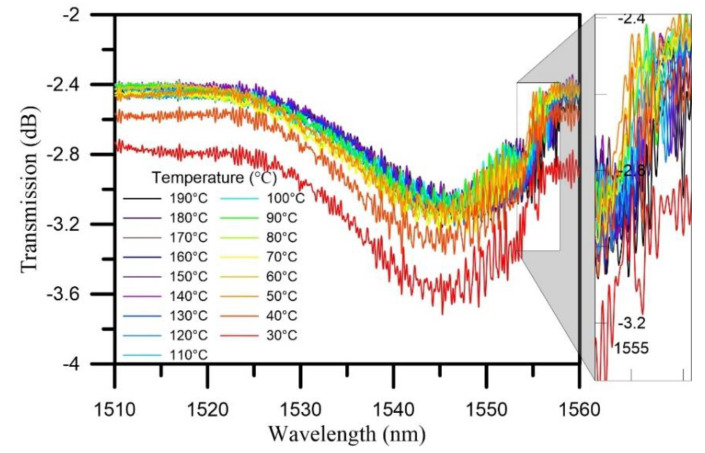
The transmission spectrum of the TFBG sensor during the cooling stage.

**Figure 11 polymers-12-01479-f011:**
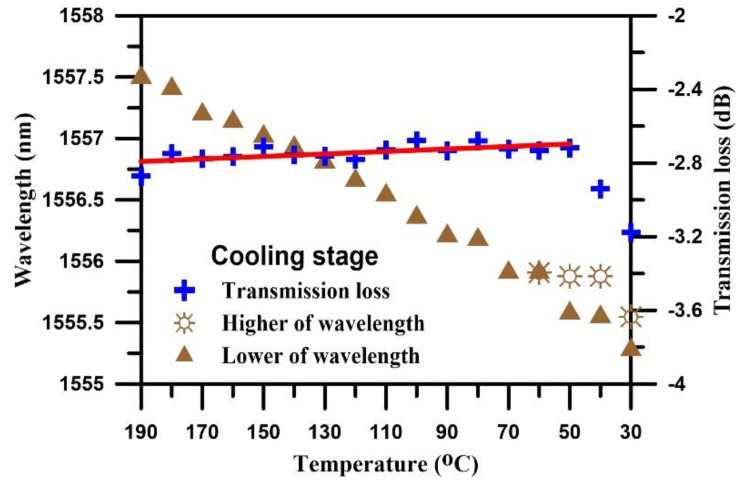
Temperature dependence of wavelength and transmission loss when cooling stage.

**Figure 12 polymers-12-01479-f012:**
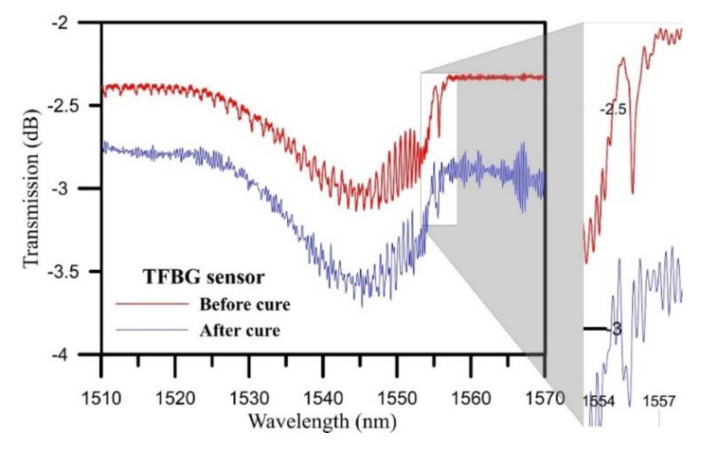
Transmission spectrum locations before and after curing.

**Table 1 polymers-12-01479-t001:** TFBG sensor residual stresses and residual strains.

TFBG Sensor		
$\sigma_{z}$	$\sigma_{x}$	$\sigma_{y}$
−22.25	−10.70	−5.27
$\varepsilon_{z}$	$\varepsilon_{x}$	$\varepsilon_{y}$
−281.351	−89.91	0

Stress (σ) unit: MPa; Strain (ε) unit: με.

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
