# Peer review of "Internal Residual Strain Measurements in Carbon Fiber-Reinforced Polymer Laminates Curing Process Using Embedded Tilted Fiber Bragg Grating Sensor"

_polymers, 2020, doi:10.3390/polym12071479_

Round 1

Reviewer 1 Report

The manuscript describes internal residual strain measurements in carbon fiber reinforced polymer laminates curing process using embedded tilted fiber Bragg grating sensor.

The paper is well-written, clear but several improvements are still necessary before publication.

Why the CFRP composite curing process presented in Figure 6 was employed?

Why the increasing and decreasing of temperature was not carried out in the same way, with the same slope temperature/time? Why 190 degree was the maximum temperature?

The transmission spectra presented in Figure 7 must be better described. What technique and instrument was employed? Please improve the experimental details.

What is the source of the noise observed in the spectra? The differences among spectra are statistically significant?

The linear fittings from Figure 8 are not really relevant. One linear fitting using 3 experimental points is not statistically correct (170, 180, 190). Please eliminate this.

In the left part of the graph, where are more experimental points the linear fitting was not included. Why?

The errors bars must be included in the figure.

In figure 9 is the similar situation as in Figure 7. Please improve this part. Both in Figure 7 and 9 please indicate the sense of increase or decrease of transmission because the colors of spectra is not really easily to follow.

In Figure 10 the connecting line among points must be deleted because there are not continuously measurements. The fittings must be statistically relevant. Please improve.

The color and thickness of legend lines from figure 10 must be improved. Why this part from the spectra was highlighted in the insert figure? All the spectra are shifted in relation with the transmittance.  Furthermore, the noise in the insert figure is greater comparing with other wavelengths. Please explain and improve.

Author Response

Thank you for your letter and for the reviewers’ comments concerning our manuscript entitled “Internal residual strain measurements in carbon fiber-reinforced polymer laminates curing process using embedded tilted fiber Bragg grating sensor” (Manuscript ID: polymers-789773). The recommendations from the reviewers were greatly appreciated and have been taken into account, as detailed in the following paragraphs, with various incomplete and inconsistent points having been revised as necessary.

The comments were all valuable and very helpful for revising and improving our paper, as well as for helping us to clarify the significance of our research. We have studied the comments carefully and have made revisions accordingly. The revised portions are marked in red in the paper. The main revisions to the paper and the related responses to the reviewers’ comments are discussed below.

Reviewer 2 Report

The manuscript "Internal residual strain measurements in carbon fiber-reinforced polymer laminates (CFRP) curing process using embedded tilted fiber Bragg grating (TFBG) sensor" are applied in this work to monitor the composite materials.

Which material applied as polymer laminates?

The content of this manuscript is unspecific as there no material characterization such as FTIR or Raman, SEM of TEM given. There need be a chapter show those. The other important point refer how a scientific work should be presented with given discussion to other work as there no references in the result part shown. This must to be included.

It also would help to give the reader more insight why such material interesting in sensor technology some sort of a general discription, such as:

Geert Luyckx, Eli Voet, Nicolas Lammens and Joris Degrieck, Strain Measurements of Composite Laminates with Embedded
Fibre Bragg Gratings: Criticism and Opportunities for Research, Sensors 2011, 11, 384-408; doi:10.3390/s110100384

Table 1 shows the strain but there no explanation where those values come from. Also there no standard deviation given so please include those.

Page 2 line 77: [0o]28, What means 28?

Page 4, line 155: the orientation shown already in Figure 3 why they need appear to be double?

If you use sensor applications there need be a calibration curve as well where the sensitivity lays. Those parts completly missing as well you show only transmission spectrum at different temperature. How did you calibrate?

Author Response

(The authors gave the same response as above.)
